# Risk of Venous Thromboembolic Events in Patients with Osteonecrosis of the Femoral Head Undergoing Primary Hip Arthroplasty

**DOI:** 10.3390/jcm8122158

**Published:** 2019-12-06

**Authors:** Pei-Hsun Sung, Yao-Hsu Yang, Hsin-Ju Chiang, John Y. Chiang, Hon-Kan Yip, Mel S. Lee

**Affiliations:** 1Division of Cardiology, Department of Internal Medicine, Kaohsiung Chang Gung Memorial Hospital and Chang Gung University, College of Medicine, Kaohsiung 83301, Taiwan; e12281@cgmh.org.tw (P.-H.S.); han.gung@msa.hinet.net (H.-K.Y.); 2Center for Shockwave Medicine and Tissue Engineering, Kaohsiung Chang Gung Memorial Hospital, Kaohsiung 83301, Taiwan; 3Department of Traditional Chinese Medicine, Chang Gung Memorial Hospital, Chiayi Branch, Putzu 61363, Taiwan; r95841012@ntu.edu.tw; 4Health Information and Epidemiology Laboratory of Chang Gung Memorial Hospital, Chiayi Branch, Putzu 61363, Taiwan; 5School of Medicine, Chang Gung University, Taoyuan 33302, Taiwan; 6Institute of Occupational Medicine and Industrial Hygiene, National Taiwan University College of Public Health, Taipei 10041, Taiwan; 7Department of Obstetrics and Gynecology, Kaohsiung Chang Gung Memorial Hospital and Chang Gung University, College of Medicine, Kaohsiung 83301, Taiwan; n22370@cgmh.org.tw; 8Chung Shan Medical University School of Medicine, Taichung 40201, Taiwan; 9Tajen University, Pingtung 90741, Taiwan; 10Department of Computer Science & Engineering, National Sun Yat-sen University, Kaohsiung 80424, Taiwan; chiang@mail.cse.nsysu.edu.tw; 11Department of Healthcare Administration and Medical Informatics, Kaohsiung Medical University, Kaohsiung 80708, Taiwan; 12Department of Medical Research, China Medical University Hospital, China Medical University, Taichung 40402, Taiwan; 13Department of Nursing, Asia University, Taichung 41354, Taiwan; 14Department of Orthopedics, Kaohsiung Chang Gung Memorial Hospital and Chang Gung University, College of Medicine, Kaohsiung 83301, Taiwan

**Keywords:** osteonecrosis of femoral head, hip replacement surgery, venous thromboembolic events, deep venous thrombosis, pulmonary embolism, risk, population-based cohort study

## Abstract

Previous data have shown patients with osteonecrosis of the femoral head (ONFH) have increased lifelong risk of unprovoked venous thromboembolic events (VTE) as compared with the general population, according to sharing common pathological mechanism of endothelial dysfunction. However, whether the risk of VTE increases in those ONFH patients undergoing major hip replacement surgery remains unclear. This is a retrospective nationwide Asian population-based study. From 1997 to 2013, a total of 12,232 ONFH patients receiving partial or total hip replacement for the first time and revision surgeries were retrospectively selected from Taiwan Health Insurance surgical files. By 1:1 matching on age, sex, surgical types, and socioeconomic status, 12,232 subjects without ONFH undergoing similar hip surgery were selected as non-ONFH group. The incidence and risk of post-surgery VTE, including deep venous thrombosis (DVT) and pulmonary embolism (PE), were compared between the ONFH and non-ONFH groups. Results showed that 53.8% of ONFH patients were male and the median age was 61.9 years old. During the mean follow-up period of 6.4 years, the incidences of VTE (1.4% vs. 1.2%), DVT (1.1% vs. 0.9%), and PE (0.4% vs. 0.4%) were slightly but insignificantly higher in the ONFH than in the non-ONFH group undergoing the same types of major hip replacement surgery (all *p*-values > 0.250). Concordantly, we found no evidence that the risk of VTE was increased in the ONFH patients as compared with the non-ONFH counterparts (adjusted HR 1.14; 95% CI 0.91–1.42; *p* = 0.262). There were also no increased risks for DVT and PE in the ONFH subgroups stratified by comorbidities, drug exposure to pain-killer or steroid, and follow-up duration after surgery, either. In conclusion, hip arthroplasty in Asian patients with ONFH is associated with similar rates of VTE as compared to patients with non-ONFH diagnoses.

## 1. Introduction

Osteonecrosis of the femoral head (ONFH) commonly affects middle-aged adults and there are more than 10,000 incidental ONFH per year reported in the United States [1]. ONFH is an avascular necrotic process of bone or marrow with mild and ambiguous symptoms in early stage and potentially disabling in late stage [2,3]. With the disease progression, the last effective therapy for advanced ONFH can only be resorted to total hip replacement [4,5]. In fact, ONFH accounts for more than 10% of total hip replacement in the United States [6] where nearby 80% of non-traumatic ONFH may attribute to steroid and alcohol use [7,8]. Especially, the relatively young victims of ONFH often need multiple replacement surgeries in their lifetime [9].

The exact pathogenesis of ONFH remains unidentifiable and is still possible to be identified in the future. The disease mechanisms of ONFH are often considered as multifactorial, including trauma, drug, surgery, and idiopathic causes [1,10,11]. Recently, growing evidence has shown that endothelial dysfunction related inappropriate nitric oxide production may be one of the key causative factors for ONFH [12,13,14,15,16]. Endothelial cell injury or dysfunction is well known as one of the fundamental risk factors for venous thromboembolic events (VTE) [17]. Undoubtedly, surgery, especially of lower extremities, substantially increases the risk of VTE [18]. Hence, anticoagulant or intermittent pneumatic compression for thromboprophylaxis in patients receiving surgery to reduce perioperative thromboembolism is recommended by clinical guidelines [19]. Compared to those with other reasons for lower limb surgery (LLS), ONFH population develops symptoms in young age, possesses diathesis of endothelial dysfunction, and may undergo several times of hip surgery in their lifetime. In a study comparing 20 ONFH patients with age-, gender-, income- and urbanization-matched population, the incidence and risk of unprovoked VTE are significantly higher in ONFH patients. However, subjects who had LLS within one year since enrollment or diagnosed with VTE within one year after surgery were excluded in that study [20]. Thus, the risk of postoperative VTE in patients with ONFH undergoing hip replacement surgery remains regrettably unanswered. In this study, we intended to investigate whether the risk of VTE after hip replacement surgery is significantly higher in patients with ONFH than those without ONFH.

## 2. Materials and Methods

### 2.1. Data Source from Taiwan National Health Insurance Research Database (NHIRD)

Healthcare is provided to near 99% of the 23.74 million Taiwanese by Taiwan National Health Insurance program [21]. The database collects information from 97% of the hospitals and clinics in Taiwan. The researchers can register and claim data of residents systematically selected from all insured enrollees of the NHIRD. The dataset included robust information regarding medical facilities, details of inpatient and outpatient orders, dental services, regimen prescription, patient care provided by physicians, and other registration files, e.g., payment, regions, and catastrophic illness, except for laboratory data and examination reports. Diagnoses are entered based on the International Classification of Diseases, 9th Revision, Clinical Modification (ICD-9-CM). This study was approved by the Ethics Institutional Review Board of Chang Gung Memorial Hospital (IRB No. 201900220B0).

### 2.2. Study Population

This was a retrospective nationwide population-based cohort study. From January 1997 to December 2013, 391,688 subjects undergoing LLS (procedure codes: 64162B, 64164B, 64169B, 64170B, 64201B, 64258B) were selected from all insured individuals in Taiwan NHIRD. LLS comprised partial and total replacements of hip or knee and revision operations. A total of 299,321 subjects who had received LLS for the first time were enrolled, because repeated surgeries would significantly increase the risk of VTE and confound the study interpretation. Subsequently, 157,394 eligible subjects undergoing partial or total hip replacement (PHR or THR, codes: 64170B, 64162B) and revision operations (64201B, 64258B) were chosen. We further divided them into the study and control groups according to the diagnosis of ONFH or not (ICD-9-CM codes: 733.42), regardless of traumatic or non-traumatic types. After excluding those with VTE diagnosed before LLS, age < 18 years old and missing baseline information, the study and control groups were matched by age, sex, income, urbanization and types of surgery in a 1:1 ratio. Finally, a total of 12,232 ONFH and 12,232 non-ONFH patients were enrolled (Figure 1). Additionally, ONFH patients undergoing primary PHR or THR for the contralateral lesion or revision operations for primary PHR or THR failure were also excluded.

Urbanization was categorized into four levels, i.e., from level 1 to 4 corresponding to the least (country) to the most urbanized level (city). Insurance taxable income level was stratified into four classifications based on monthly insurance payment of individual insured enrollee, i.e., level 1: none, level 2, 1–15,840, level 3: 15,841–25,000, and level 4: >25,000 New Taiwan Dollars per month. These two background parameters were matched between groups to eliminate the baseline differences in patients’ lifestyle and socioeconomic status.

### 2.3. Diagnosis Confirmation and Acquisition of Medical and Surgical Information

Diagnostic data of VTE, including deep venous thrombosis (DVT; ICD-9-CM codes 453.8) and pulmonary embolism (PE; 415.1), were retrieved from NHIRD based upon entered clinical or hospitalization information [22]. The diagnoses were documented by at least three records of consecutive outpatient visits within one year or one proved diagnosis on admission during study period. We also checked the corresponding thrombolytic agents or anticoagulants to confirm these venous thromboembolic diagnoses. Additionally, drug exposure to non-steroidal anti-inflammatory drug (NSAID) or systemic corticosteroid therapy (steroid) was defined as the regimen prescription duration for ≥1 month. In contrast, we defined those without prescription or short-term (<1 month) use of the above two medications as non-drug exposure, because they might be prescribed for other purposes, rather than standard treatment or symptomatic relief for ONFH.

Data of surgical procedures were retrieved from Taiwan Health Insurance surgical files in which total patients’ information of the performed orthopedic surgical procedures were registered. The relevant diagnoses, surgical types, date, and locations were acquired for baseline matching and further analysis. The first day of surgery was defined as surgical index date. Those with missing or incomplete data entered were excluded from the study.

### 2.4. Comorbidities and Outcomes

The first day since the patient’s enrollment was defined as the index date. During the 17-year dataset period, we evaluated the comorbidities including traditional atherosclerotic risk factors, relevant cardiovascular or non-cardiovascular systemic diseases for each subject followed. Diagnoses of comorbid diseases were confirmed by the same three consecutively ICD-9-CM codes, such as hypertension (ICD-9-CM codes 401-405), diabetes (250), dyslipidemia (272), gout (274), systemic lupus erythematosus (710.0), atrial fibrillation (427.31), chronic ischemic heart disease (412–414, 429.2), peripheral vascular disease (440, 443.9,444.0, 444.2, 444.8, 444.9, 447.8, 447.9, 445.0, 445.02), chronic kidney disease (585), and obesity (278). The frequency of the above comorbidities in these two groups during the whole study period were evaluated and compared.

The first day of clinical events, i.e., VTE, DVT, or PE, was defined as event date and was censored as the end of study. We aimed to evaluate the frequency and incidence of VTE after PHR, THR, or revision operation in both ONFH and non-ONFH groups. Additionally, the risk of VTE after hip surgery for ONFH was analyzed to clarify whether the risk is higher in the ONFH than non-ONFH group. Furthermore, whether the risk changes with duration of drug exposure or time period after surgery was also clarified.

### 2.5. Statistical Analysis

After matching with age, sex, socioeconomic status, and types of surgery, the demographic data and frequency of comorbidities and outcomes between the ONFH and matched non-ONFH cohorts were compared with the independent t and Chi-square tests as appropriate. The incidence rate and 95% confidence interval (CI) of VTE, including DVT and PE, were calculated between groups in the entire follow-up period. Additionally, we used the Kaplan-Meier method to estimate cumulative incidences in both groups and performed the Log-rank test to examine differences between them. By using Cox proportional hazard regression model, risks of ONFH for VTE, DVT, and PE were analyzed and the data was expressed with the hazard ratio (HR) concomitant with 95% CI for each parameter. Multivariate analysis was adjusted for background parameters including age, sex, urbanization, and income levels, and drug exposure under the circumstance of matched surgical procedures. Two-tailed *p*-value < 0.05 was considered statistically significant. All analyses were conducted using SAS statistical software (Version 9.4; SAS Institute, Cary, NC, USA).

## 3. Results

### 3.1. Demographic Data of the Surgical Patients with and without ONFH

Table 1 shows that more than one half of ONFH patients were male and underwent first hip surgery before 65 years old, with the median age about 62 years old. Only a minority of them had a poor socioeconomic status. About 70% and 30% of ONFH cases received THR and PHR, respectively, at the first time of hip surgery. Even though we excluded those with repeated LLS in the beginning, there were still 1% of cases undergoing revision operation, suggesting their first primary THR or PHR was performed before 18 years ago. Additionally, the surgical ONFH patients had significantly higher frequencies of dyslipidemia, gout, and systemic lupus erythematosus (all *p* < 0.002). By contrast, non-ONFH population had significantly more prevalence of diabetes and obesity (all *p* < 0.02). Of note, one third of ONFH patients had concomitant coronary heart disease. The frequency of drug exposure to NSAID or steroid for ≥1 month was significantly higher in the ONFH than non-ONFH group (all *p* < 0.001). More than 80% of ONFH patients still needed long-term use of NSAID for pain relief after surgery.

### 3.2. Outcome Comparison for Incidence Rate and Adjusted Risk of VTE, DVT, and PE

During the mean follow-up period of 6.4 years, the ONFH patients undergoing hip surgery did not have significantly higher frequencies of VTE, DVT, and PE compared with the non-ONFH group (all *p* = NS) (ref. the bottom of Table 1). Table 2 displays the incidence rate of VTE was 21.2 and 19.4 per 100,000 person-years in the ONFH and non-ONFH group, respectively. Therefore, there was no evidence that the surgical ONFH patients had an increased risk for incidental VTE compared to those without ONFH given the statistical result (95% CI 0.88–1.36; *p* = 0.440). Also, the incidences of DVT and PE displayed the similar insignificant pattern to that of VTE. Regarding occurrence of VTE in relation to time period since first hip surgery for ONFH, the Kaplan-Meier curve in Figure 2 demonstrates that there were similar cumulative incidences of VTE, including DVT and PE, among the ONFH and non-ONFH groups in the 17-year study period (all *p* = NS with Log-rank test).

Stratified analysis on Table 2 demonstrates the risk of VTE in ONFH was essentially invariant with not only the duration of drug exposure to NSAID and steroid, but also the time period after hip surgery, suggesting there was no short-term or long-term concern about increased risk of postoperative VTE in the ONFH population receiving major hip replacement surgery.

### 3.3. Multivariate Analysis for Identifying the Risk Factors for Post-Operative VTE, DVT and PE

As shown in Table 3, the results of Cox regression analysis adjusted for age, sex, socioeconomical background, comorbidities, and drugs revealed that advanced age was significantly associated with increased risk of VTE, DVT, and PE. On the contrary, short- or long-term use of NSAID was incidentally found to be protective against VTE. The interesting finding can be attributed to that early/normal ambulation after hip surgery by adequate pain control might be helpful to prevent attack of VTE. Additionally, other underlying diseases, aside from ONFH, were found not to increase the risk of VTE after surgery. Again, steroid use, which was an independent risk factor for ONFH and strongly associated with endothelial dysfunction, also did not increase VTE risk.

## 4. Discussion

The present study using Taiwan NHIRD to compare the risk of post-operative VTE between ONFH and matched non-ONFH patients yielded several novel findings and provided useful clinical information. Firstly, the incidences of VTE, DVT, and PE were slightly but insignificantly higher in the ONFH than non-ONFH group undergoing the same types of hip surgery. Secondly, the surgical ONFH patients did not have a significantly increased risk of VTE as compared with surgical non-ONFH counterparts. Thirdly, there were no increased risks for VTE in the ONFH subgroups stratified by comorbidities, durations of drug exposure to NSAID or steroid, and follow-up periods after surgery. These findings from an Asian population-based study implied that although ONFH patients have underlying pathological mechanism of endothelial dysfunction, they did not have higher postoperative risk of VTE, including DVT and PE, as compared to those without ONFH. As a result, we suggested hip replacement surgery could be equally safe in the ONFH population without an extra need of increased dose or duration of prophylactic oral anticoagulant in terms of an increased risk for surgery-related VTE.

Two previous clinical observational researches to investigate the association between ONFH and cardiovascular diseases [14,20] have shown positive results that both major adverse cardiovascular or cerebrovascular events (MACCE) and VTE were significantly increased in the ONFH population as compared with the general population. The underlying mechanism is thought to be related to the endothelial dysfunction. In the former study [14], the ONFH patients were found to have about twice increased risk for MACCE even after multivariate adjustment for medications, surgery and relevant comorbidities. The latter study was aimed to find the incidence of unprovoked VTE in non-traumatic ONFH [20]. Those subjects with precedent trauma, underwent LLS in one year, and occurrence of VTE within one year after surgery were excluded, due to an obviously increased VTE risk through blood stasis and vascular/endothelial injury, the two crucial components of Virchow’s triad [23]. Notably, nontraumatic ONFH was found to be doubly risky for unprovoked VTE as compared with general population. However, whether ONFH patients, either traumatic or nontraumatic, undergoing major hip replacement surgery would have higher post-operative VTE risk compared to matched control group remains unanswered. 

In the previous study, the incidences of unprovoked VTE, DVT, and PE were 1.2%, 1.0%, and 0.2%, respectively [20]. Current study found the corresponding incidences were slightly increased to 1.4%, 1.1%, and 0.4%, respectively. These results are reasonable because the extrinsic pathway of coagulation cascade triggered by surgery may only have marginal effect and plateaued on the risk of VTE when the intrinsic pathway in ONFH patients has already been primed [24]. Nevertheless, while comparing to another data also from Asia showing that asymptomatic thrombophlebitis diagnosed with ultrasound is about 5% [25], we noted that VTE after hip surgery in Taiwan was quite low, i.e., only around 1%, even without anticoagulant prevention. Therefore, we were not sure whether the result could become statistically significant if asymptomatic VTE were detected by ultrasound or other image studies. It is also possible that the incidence of VTE in our study might be underestimated due to the inherent limitations of this administrative database. Furthermore, the rate of VTE in some western populations is around 10% in absence of thromboprophylaxis, and the rate of asymptomatic thrombophlebitis is even up to 20% [26]. Taken together, application of the results from our Asian-based study for non-Asian populations needs careful interpretation and more investigation. 

Short-term or long-term administration of NSAID protected ONFH patients from postoperative VTE was noted in the present study. Also, data from the previous two studies [14,20] have documented that NSAID has the protective effects against not only provoked VTE but also MACCE in patients with nontraumatic ONFH. Although most early researches have shown the use of non-selective NSAID strongly leads to endothelial dysfunction and links to arterial/venous thrombotic complications [27,28], yet more and more recent meta-analyses have challenged the cardiovascular safety issue of NSAID, especially cyclooxygenase-2 (COX-2) selective inhibitor [29,30]. Our findings, corresponding to less safety concern about thrombogenic property of NSAID in the recent reviews, supported the use of NSAID transiently or regularly for pain relief of ONFH not only did not increase the both risks of unprovoked VTE and MACCE, but also protected against surgery-related VTE. Adequate pain control was supposed to facilitate early ambulation in those surgical ONFH patients, and then further prevent thromboembolic events. 

We did not find an increased postoperative risk of VTE in Asian ONFH population in the present study. The analytical results were consistent after stratified with age, gender, comorbidities, duration of drug exposure and postoperative period. Furthermore, the incidence of VTE in ONFH after surgery was about 1.4% and the risk did not increase within 1 month or even beyond 5 years after surgery. Therefore, based upon the present findings, titration up of anticoagulant dose or extension of anticoagulant duration for enhancement of thromboprophylaxis might be not suggested for Asian ONFH patients who require surgery or have received an operation. At least, we can safely follow standard thromboprophylactic strategies recommended by current guidelines [19].

There were some study limitations. First, detailed personal history and lifestyle information such as smoking, alcohol, body mass index, and life independence are not provided by Taiwan NHIRD. Second, all data have been registered with ICD-9-CM codes, and therefore further classification of disease severity was impracticable. The possibility of misclassification bias might also be considered. Third, the laboratory data and detailed examination reports are not available in NHIRD, so all diagnostic precisions depend solely upon entered data. Fourth, pharmacological or mechanical thromboprophylaxis was not routinely adopted in preoperative setting in Taiwan, and therefore we did not analyze the impact of preoperative anticoagulants on the risk of VTE between groups. Fifth, there was no information regarding over-the-counter medication use. Finally, NHIRD is a nationwide database of Asian population. Although the prevalence and incidence of VTE were compatible with the systematically reviewed data [31], ethnic difference shall be taken into account while applying the study results into clinical practice globally.

## 5. Conclusions

The risk of VTE, including DVT and PE, after major hip replacement surgery was similar among the Asian patients with and without ONFH. Further research in this area is needed to confirm whether titration up of anticoagulant dose or extension of anticoagulant duration is warranted in this population.

## Figures and Tables

**Figure 1 jcm-08-02158-f001:**
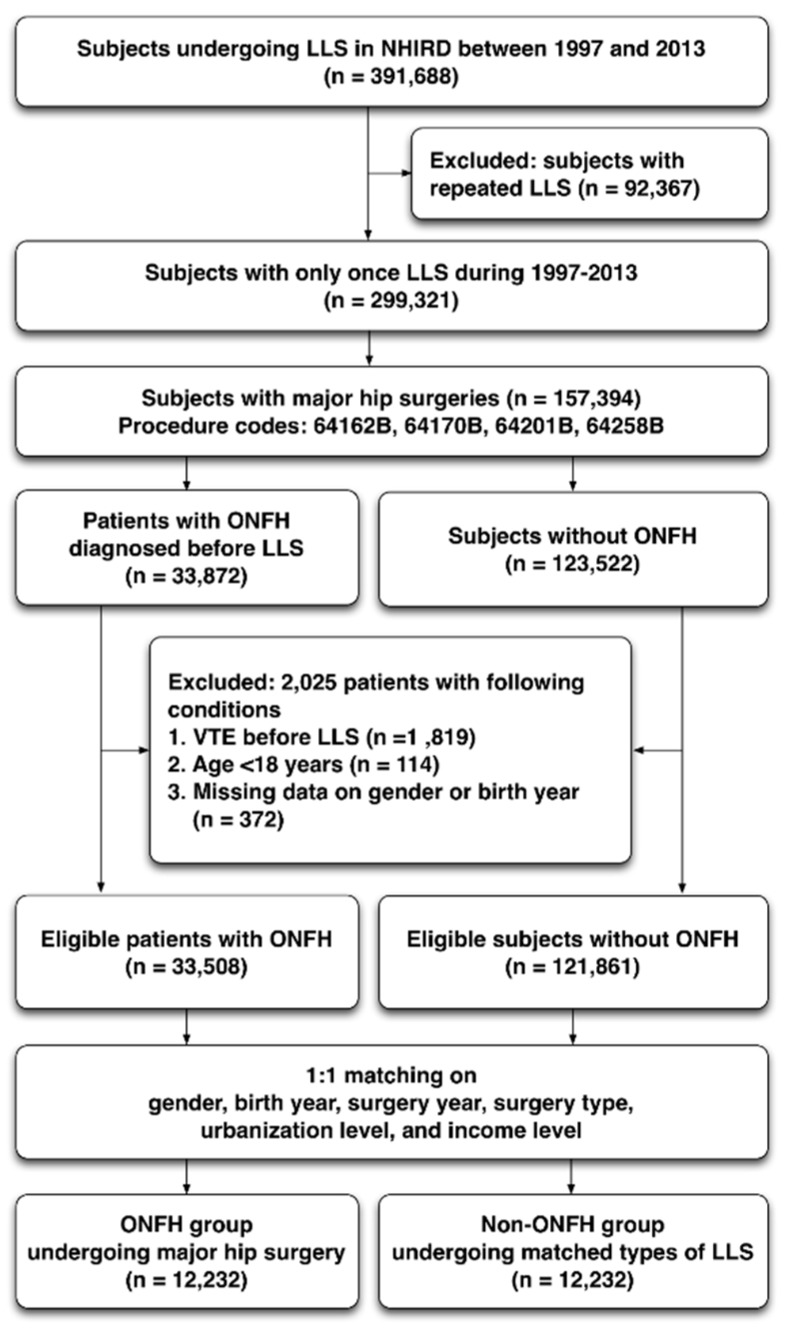
Flow diagram regarding how eligible patients were selected and allocated into the ONFH and matched non-ONFH groups. Non-ONFH group undergoing hip surgery was selected by matching ONFH group with age, sex, socioeconomic status, and types of surgery in a 1:1 ratio after excluding repeated surgeries, VTE before LLS, age < 18 years old, and incomplete or missing detailed information. Abbreviation: NHIRD—National Health Insurance Research Database; LLS—lower limb surgery; ONFH—osteonecrosis of femoral head; VTE—venous thromboembolic events.

**Figure 2 jcm-08-02158-f002:**
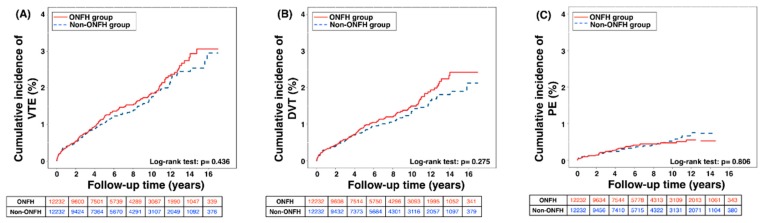
Cumulative incidence of (**A**) VTE, (**B**) DVT, and (**C**) PE in the surgical patients with and without ONFH in 17-year dataset period. Abbreviation: VTE—venous thromboembolic events; DVT—deep vein thrombosis; PE—pulmonary embolism; ONFH—osteonecrosis of femoral head.

**Table 1 jcm-08-02158-t001:** Demographic characteristics and outcome of VTE in the surgical patients with and without ONFH, matched by any type of hip surgery.

	ONFH Group (*n* = 12232)		Non-ONFH * Group (*n* = 12232)		
Variables	No.	%	No.	%	*p*-Value
Gender					1.000
Female	5654	46.2	5654	46.2	
Male	6578	53.8	6578	53.8	
Age (years)					1.000
18–65	6533	53.4	6533	53.4	
>65	5699	46.6	5699	46.6	
Median age (IQR)	61.9 (53–72)		61.9 (53–72)		
Urbanization level					1.000
1 (rural)	1212	9.9	1212	9.9	
2	2441	20.0	2441	20.0	
3	5990	49.0	5990	49.0	
4 (urban)	2589	21.2	2589	21.2	
Income level					1.000
1 (lowest)	3319	27.1	3319	27.1	
2	2075	17.0	2075	17.0	
3	4896	40.0	4896	40.0	
4 (highest)	1942	15.9	1942	15.9	
Surgical procedure					1.000
THR	8516	69.6	8516	69.6	
PHR	3596	29.4	3596	29.4	
Revision of hip replacement	120	1.0	120	1.0	
Comorbidities					
Hypertension	7694	62.9	7836	64.1	0.059
Diabetes mellitus	3592	29.4	3902	31.9	<0.001
Dyslipidemia	4597	37.6	4350	35.6	0.001
Gout	3138	25.7	2658	21.7	<0.001
Systemic lupus erythematosus	214	1.7	61	0.5	<0.001
Atrial fibrillation	649	5.3	622	5.1	0.437
Coronary heart disease	3836	31.4	3734	30.5	0.158
Peripheral vascular disease	1010	8.3	1048	8.6	0.381
Chronic kidney disease	1217	9.9	1248	10.2	0.510
Obesity	116	0.9	156	1.3	0.015
Drug exposure (>1 month)					
NSAID	10,242	83.7	9623	78.7	<0.001
Steroid	4099	32.8	3255	26.6	<0.001
Outcomes					
VTE	167	1.4	152	1.2	0.398
DVT	133	1.1	115	0.9	0.251
PE	44	0.4	46	0.4	0.833

* Control group (non-ONFH group) was matched by age, gender, income/urbanization level, and types of hip surgery. Abbreviations: VTE—venous thromboembolic event; ONFH—osteonecrosis of femoral head; No.—number; IQR—interquartile range; THR—total hip replacement; PHR—partial hip replacement; NSAID—non-steroid anti-inflammatory drug; DVT—deep venous thrombosis; PE—pulmonary embolism.

**Table 2 jcm-08-02158-t002:** Stratified comparison of incidence rate and risk of VTE between the surgical patients with and without ONFH.

	ONFH (*n* = 12232)			Non-ONFH (*n* = 12232)					
Variables	Event	PY	Rate *	Event	PY	Rate	IRR	95% CI	*p*-Value
VTE	167	78,820.55	21.19	152	78,224.69	19.43	1.09	0.88–1.36	0.440
DVT	133	78,940.20	16.85	115	78,356.04	14.68	1.15	0.89–1.47	0.279
PE	44	79,253.03	5.55	46	78,669.39	5.85	0.95	0.63–1.44	0.806
Gender									
Female	101	34,524.82	29.25	62	34,457.93	17.99	1.63	1.19–2.23	0.003
Male	66	44,295.73	14.90	90	43,766.76	20.56	0.72	0.53–1.00	0.047
Age									
18–65 years	58	42,933.20	13.51	64	42,372.27	15.10	0.89	0.63–1.28	0.538
>65 years	109	35,887.35	30.37	88	35,852.42	24.55	1.24	0.93–1.64	0.137
Hypertension									
No	43	28,179.04	15.26	48	27,305.24	17.58	0.87	0.58–1.31	0.500
Yes	124	50,641.50	24.49	104	50,919.45	20.42	1.20	0.92–1.56	0.173
Diabetes mellitus									
No	105	54,228.76	19.36	104	51,999.40	20.00	0.97	0.74–1.27	0.815
Yes	62	24,591.79	25.21	48	26,225.29	18.30	1.38	0.95–2.01	0.096
Dyslipidemia									
No	93	49,459.34	18.80	103	51,513.41	19.99	0.94	0.71–1.24	0.668
Yes	74	29,361.21	25.20	49	26,711.28	18.34	1.37	0.96–1.97	0.085
Gout									
No	118	57,675.98	20.46	115	60,791.02	18.92	1.08	0.84–1.40	0.550
Yes	49	21,144.57	23.17	37	17,433.67	21.22	1.09	0.71–1.67	0.686
Systemic lupus erythematosus									
No	163	77,577.48	21.01	150	77,867.36	19.26	1.09	0.87–1.36	0.443
Yes	4	1243.06	32.18	2	357.33	55.97	0.57	0.11–3.14	0.523
Atrial fibrillation									
No	154	74,240.54	20.74	139	73,796.08	18.84	1.10	0.88–1.39	0.410
Yes	13	4580.01	28.38	13	4428.61	29.35	0.97	0.45–2.09	0.932
Chronic ischemic heart disease									
No	85	52,647.86	16.15	96	53,130.35	18.07	0.89	0.67–1.20	0.450
Yes	82	26,172.68	31.33	56	25,094.34	22.32	1.40	1.00–1.97	0.050
Peripheral vascular disease									
No	140	71,858.15	19.48	132	71,302.58	18.51	1.05	0.83–1.33	0.674
Yes	27	6962.40	38.78	20	6922.11	28.89	1.34	0.75–2.39	0.319
Chronic kidney disease									
No	146	70,479.76	20.72	133	70,274.14	18.93	1.09	0.87–1.38	0.451
Yes	21	8340.79	25.18	19	7950.55	23.90	1.05	0.57–1.96	0.869
Obesity									
No	166	78,215.13	21.22	150	77,406.90	19.38	1.10	0.88–1.37	0.419
Yes	1	605.42	16.52	2	817.79	24.46	0.68	0.06–7.45	0.749
Exposure to NSAID									
<1 month	30	6941.74	43.22	39	10,403.76	37.49	1.15	0.72–1.86	0.558
1–6 months	47	24,723.47	19.01	50	25,783.44	19.39	0.98	0.66–1.46	0.922
>6 months	90	47,155.33	19.09	63	42,037.49	14.99	1.27	0.92–1.76	0.141
Exposure to steroid									
<1 month	99	46,136.23	21.46	105	51,630.36	20.34	1.06	0.80–1.39	0.702
1–6 months	45	21,297.60	21.13	32	18,851.76	16.97	1.24	0.79–1.96	0.344
>6 months	23	11,386.72	20.20	15	7742.57	19.37	1.04	0.54–2.00	0.900
Follow-up period after surgery									
<30 days	9	999.91	90.01	12	1000.38	119.95	0.75	0.32–1.78	0.515
31–365 days	34	10,464.17	32.49	34	10,380.73	32.75	0.99	0.62–1.60	0.974
1–3 years	38	19,209.97	19.78	31	18,864.33	16.43	1.20	0.75–1.93	0.444
3–5 years	32	15,023.24	21.30	22	14,751.51	14.91	1.43	0.83–2.46	0.198
>5 years	54	33,123.25	16.30	53	33,227.75	15.95	1.02	0.70–1.49	0.910

* Rate denotes incidence rate (per 10,000 person-years). Abbreviations: VTE—venous thromboembolic event; DVT—deep venous thrombosis; PE—pulmonary embolism; ONFH—osteonecrosis of femoral head; PY—person-years; IRR—incidence rate ratio; CI—confidence interval; NSAID—non-steroid anti-inflammatory drug.

**Table 3 jcm-08-02158-t003:** Multivariate * Cox regression analysis for identifying the predictors of VTE, including DVT and PE, after hip replacement surgery.

	VTE			DVT			PE		
	aHR	95% CI	*p*-Value	aHR	95% CI	*p*-Value	aHR	95% CI	*p*-Value
ONFH	1.14	0.91–1.42	0.262	1.22	0.94–1.56	0.130	0.95	0.62–1.44	0.802
Gender (ref.: female)									
Male	0.84	0.66–1.07	0.152	0.92	0.69–1.21	0.530	0.68	0.43–1.09	0.106
Age (ref.: 18–65 years)									
>65 years	1.62	1.24–2.11	<0.001	1.55	1.15–2.10	0.004	1.72	1.04–2.84	0.035
Comorbidities									
Hypertension	1.09	0.82–1.45	0.540	1.11	0.81–1.53	0.507	1.01	0.59–1.73	0.978
Diabetes mellitus	0.97	0.76–1.24	0.796	1.06	0.80–1.41	0.678	0.73	0.45–1.17	0.185
Dyslipidemia	1.05	0.81–1.35	0.732	0.94	0.70–1.25	0.649	1.58	0.99–2.51	0.053
Gout	1.23	0.94–1.60	0.133	1.26	0.93–1.70	0.130	1.05	0.63–1.75	0.861
Systemic lupus erythematosus	2.00	0.87–4.58	0.104	1.83	0.67–5.05	0.242	1.99	0.47–8.48	0.354
Atrial fibrillation	1.12	0.74–1.70	0.580	1.08	0.67–1.75	0.748	1.34	0.66-2.73	0.419
Chronic ischemic heart disease	1.36	1.05–1.74	0.018	1.23	0.92–1.65	0.155	1.71	1.07–2.73	0.025
Peripheral vascular disease	1.57	1.14–2.17	0.006	1.96	1.39–2.78	<.001	0.90	0.44–1.82	0.763
Chronic kidney disease	0.96	0.68–1.35	0.810	1.04	0.71–1.52	0.859	0.65	0.31–1.37	0.254
Obesity	1.06	0.34–3.32	0.924	0.46	0.06–3.29	0.439	2.36	0.57–9.76	0.236
Drug exposure									
NSAID (ref.: <1 month)									
1–6 months	0.48	0.35–0.66	<0.001	0.50	0.35–0.71	<0.001	0.51	0.28–0.92	0.027
>6 months	0.37	0.27–0.51	<0.001	0.37	0.26–0.53	<0.001	0.34	0.18–0.63	0.001
Steroid (ref.: <1 month)									
1–6 months	1.05	0.80–1.39	0.719	0.97	0.70–1.34	0.859	1.38	0.82–2.32	0.231
>6 months	1.03	0.71–1.49	0.886	0.86	0.56–1.35	0.520	1.83	0.97–3.43	0.062

* Multivariate analysis was adjusted for age, gender, level of urbanization/income, comorbidities, and drugs. Abbreviations: VTE—venous thromboembolic event; DVT—deep venous thrombosis; PE—pulmonary embolism; ref.—reference; aHR—adjusted hazard ratio; CI—confidence interval; ONFH—osteonecrosis of femoral head; NSAID—non-steroid anti-inflammatory drug.

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
