# Peer review of "Risk of Venous Thromboembolic Events in Patients with Osteonecrosis of the Femoral Head Undergoing Primary Hip Arthroplasty"

_jcm, 2019, doi:10.3390/jcm8122158_

Round 1

Reviewer 1 Report

The authors have answered the minor objections stated in the review and have corrected the manuscript sufficiently.

Reviewer 2 Report

The subject is according to the scope of the Journal and the chosen topic is of scientific interest.

The manuscript is well organized and logical and could be accepted for publication.

Reviewer 3 Report

Congratulations. Well done and presented. It will be interesting to dedicate a next study to comparison PHR and CHR in the same aspects ok?

This manuscript is a resubmission of an earlier submission. The following is a list of the peer review reports and author responses from that submission.

Round 1

Reviewer 1 Report

the study is an excellent study which is well powered and studies an important area of DVT and treatment.  It is important to seperate the area and risk adjust of  patients.  The manuscripts adds to literature to supports the ON in the asian population

Reviewer 2 Report

The aim of this study was to answer the question if patients with osteonecrosis of the femoral head suffer higher risks of venous thromboembolic events after hip replacement surgery.

The manuscript describes an obviously very well conducted study at a high technical and methodological level.

The results are clinically relevant, concern large numbers of patients and provide pertinent and valuable recommendations for treatment. The presentation of results and the discussion are adequate in content and length. I have therefore the seldom pleasure to recommend publication without substantial revisions and congratulate the authors to their fine efforts.

A very few objections to be raised concern some minor inconstancies in phrasing and a few typos.

Line 72 replace “growing evidences” by “growing evidence”

Line 76: replace “especially over lower extremities” by “especially of the lower extremities”

Line 104: probably better “A total of 299,321 subjects which had received LLS only for the first time were enrolled,…”

Line 181: replace “frequency” by “frequencies”

Line 278: We did not find …

Line 279: study

Line 279: were consistent after being stratified with

Line s 180-181: Please clarify the following. The sentence “Besides, the surgical ONFH patients had significantly higher frequency of diabetes, dyslipidemia, gout, lupus and obesity…” seems to refer in context to table 1. But in this table the values for hypertension and diabetes mellitus are higher for the Non-ONFH group